# The Effects of Omega-3 Supplementation on Depression in Adults with Cardiometabolic Disease: A Systematic Review of Randomised Control Trials

**DOI:** 10.3390/nu14091827

**Published:** 2022-04-27

**Authors:** Franciskos Arsenyadis, Ehtasham Ahmad, Emma Redman, Thomas Yates, Melanie Davies, Kamlesh Khunti

**Affiliations:** 1NIHR Leicester Biomedical Research Centre, Leicester General Hospital, College of Life Sciences, University of Leicester, Leicester LE5 4PW, UK; ea306@leicester.ac.uk (E.A.); emma.redman@uhl-tr.nhs.uk (E.R.); ty20@leicester.ac.uk (T.Y.); melanie.davies@uhl-tr.nhs.uk (M.D.); kk22@leicester.ac.uk (K.K.); 2Diabetes Research Centre, University Hospitals of Leicester NHS Trust, Leicester LE5 4PW, UK; 3NIHR Applied Health Research Collaboration-East Midlands (NIHR ARC-EM), University of Leicester, Leicester LE5 4PW, UK

**Keywords:** Omega-3, polyunsaturated fat, cardiometabolic disease, depression, review

## Abstract

Background: Omega-3 polyunsaturated fatty acids′ concurrent benefits for cardiometabolic and mental health are equivocal. Despite lack of evidence, up to a third of adults consume Omega-3 supplements. No review has yet been published to report effect on depression in this cardiometabolic population. Methods: We conducted a systematic review of double-blinded, controlled randomised trials to investigate the safety and effect of Omega-3 supplementation on depression scores in people with cardiometabolic diseases. Primary outcome was change in depression scores versus placebo. Secondary outcomes were side-effects, concurrent medication and adherence. Results: Seven trials reporting on 2575 (672 female) adults aged 39–73 were included. Omega-3 dosages ranged from 1–3 g with an intervention duration of 10–48 weeks. Six out of seven trials found no statistically or clinically significant change to depression scores compared to placebo. One trial favoured intervention (Relative Risk Reduction: 47.93%, 95% CI: 24.89–63.98%, *p* < 0.001). Sub-analyses showed clinically meaningful reductions in depression scores for those on antidepressants (Intervention: 20.9 (SD: 7.1), Placebo: 24.9 (SD: 8.5) *p* < 0.05) or with severe depression (−1.74; 95% CI −3.04 to −0.05, *p* < 0.05) in two separate trials. Side effects were comparable between treatment arms. Conclusions: Omega-3 supplementation is safe to use but not superior to placebo for depression in adults with concurrent cardiometabolic disease.

## 1. Introduction

Prevalence of cardiometabolic disease, an umbrella term used to group common yet preventable non-communicable conditions such as type 2 diabetes and cardiovascular disease, is increasing globally [1]. Cardiometabolic disease is the leading cause of premature mortality and is a major contributor to the global burden of long-term conditions and multimorbidity [2]. Co-morbid mental health conditions, particularly depression, are common in people with cardiometabolic diseases. Depression is the second leading cause of years lost to disability worldwide [3] and affects an estimated 280 million people (3.8% of the world population) [4]. Depression accounts for 7.5% of all years lived with disability [5] and individuals affected by depression may experience impaired psychosocial functioning and are 11 times more likely to attempt suicide [6].

A bi-directional relationship exists between depression and cardiometabolic disease [7,8]. Depression doubles the risk of developing cardiovascular disease[9], triples mortality risk following acute myocardial infarction (AMI) and acts as a significant predictor of mortality in chronic heart failure [10,11]. However, cardiometabolic disease may also lead to depression. Up to 30% of AMI survivors develop depressive symptoms, while people with diabetes have double the rates of depression compared to the general population [12,13,14]. Depression is also associated with poor medication adherence and can reduce participation in and adherence to lifestyle interventions used in settings such as cardiac rehabilitation [15,16,17].

There are few treatment options for concurrent management of cardiometabolic disease and comorbid depression. Agents used to treat depression may negatively impact the management of cardiometabolic disease. For example, there is evidence linking use of antidepressants with increased risk of major adverse cardiovascular events and weight gain [18,19].

A potential alternative to pharmacotherapy is lifestyle intervention. Dietary approaches higher in nutrients such as Omega-3 polyunsaturated fatty acid (PUFA) and fibre (e.g., Mediterranean diet) have been associated with reduced incidence of cardiometabolic disease and depression [20]. Health behaviour interventions for people living with depression that adapt their diet have produced positive results. The SMILES trial (Australia) randomly assigned adults with moderate to severe depression to a Mediterranean-style diet or standard therapy (control) [21]. At 12 weeks, depression remission was achieved for 32.3% of those following a Mediterranean-style diet, versus 8% for control, with evidence of a dose response relationship, indicating that greater adherence to the diet was associated with better outcomes. However, there are limitations to implementing dietary interventions with free-living individuals. Long-term adherence is challenging once intensive one-to-one support ceases [22].

Prevalence of concurrent mental and cardiometabolic conditions, counterproductive treatments and potential for health behaviour adaption all highlight the need to identify complementary management strategies.

Omega-3s PUFAs are found in the human diet, in the form of alpha-linolenic acid (18: 3n-3) (ALA), eicosapentaenoic acid (20: 5n-3) (EPA), and docosahexaenoic acid (22: 6n-3) (DHA). ALA is classed as an essential Omega-3 in that it cannot be endogenously synthesised and is found in plant-based sources such as nuts and seeds. EPA can be synthesised from ALA or obtained through eating oily fish, and DHA can be synthesised from both ALA and EPA at variable rates and is also found in oily fish [23].

Omega-3s have cardiovascular benefits, likely to be linked to their triglyceride and blood pressure lowering effects [24]. The REDUCE-IT trial found that ingesting 4 g of icosapent ethyl (a purified version of EPA) daily reduced a composite of cardiovascular events, such as cardiovascular death, nonfatal myocardial infarction and nonfatal stroke in 8179 people treated for primary and secondary cardiovascular prevention [25]. Confidence in these results, however, has been questioned, as other Omega-3 supplementation trials, such as the STRENGTH, ASCEND, ORIGIN and VITAL trials, did not report any benefits [26,27,28,29].

Omega-3 supplementation for depression has already been investigated in multiple studies of adults with [30,31] or without comorbid disease [32] or pregnancy [33] and found to arrive at an overall significant reduction in depression scores [34]. This relationship, however, has not been investigated in adults with cardiometabolic disease, where Omega-3 supplementation may already have a use in primary and secondary cardiovascular disease prevention [25].

In view of the uncertainties, and to minimise biases occurring from open label trials and cohort studies, we conducted a systematic review of double-blind randomised controlled trials (RCTs) to investigate the safety, tolerability and effectiveness of Omega-3 supplementation on depression in people with cardiometabolic diseases.

## 2. Materials and Methods

The Preferred Reporting Items for Systematic Reviews and Meta-analysis (PRISMA) and Cochrane Collaboration guidance on systematic reviews was used in the development and report of the systematic review. Registration with the international prospective register of systematic reviews (PROSPERO) was obtained in advance of search strategy execution (CRD42021249222).

### 2.1. Inclusion Criteria

A preliminary scoping review informed the search strategy and database choice. This systematic review focused on the highest quality research methodology and reporting, clinical trials, with eligible studies limited to double-blind RCTs only. Observational and cohort studies, case series or open-label studies were excluded. Eligible populations were limited to adults (aged 18 years and over) with a confirmed diagnosis of cardiovascular disease, stroke and/or diabetes, or within the umbrella term of cardiometabolic disease.

The intervention eligibility was limited to Omega-3 provided in supplement form (i.e., capsule, tablet or liquid), or supplement plus diet, but excludes dietary only. Supplements had to be described as containing EPA, DHA or ALA. No further limitation was placed on formulation, dosage, duration or frequency (i.e., daily or weekly).

For the control (comparator) group, eligible studies include placebo, no intervention or standard practice for the treatment of depression (i.e., talking therapy, cognitive behavioural therapy and/or antidepressants).

The primary outcome measures were depression scores validated for use in the specific populations studied (professionally or self-assessed). Studies using self-reported depression scores using verbal or non-validated tools were excluded. Secondary outcomes include attrition, adherence and adverse events in intervention and placebo groups where available.

### 2.2. Search Strategy

Search terms include: (1) “Depression” and (2) “Omega-3”, with synonyms, common variations and sub-categories. Trial filters were applied where applicable to limit search results to relevant study designs only. The full breakdown of the search terms for each individual database (CINAHL, Medline and Cochrane Library) is uploaded in unaltered form on PROSPERO (CRD42021249222) and is provided in the Appendix A. No limits were placed on year of publication and databases were searched from year of inception. The search was limited to English language only. Grey literature and clinical trial registries of prospective trials were not searched, as they did not meet the aim of the review, which was to characterise the current peer-reviewed evidence base for double blind RCTs.

### 2.3. Electronic Searches

Database searches up to January 2022 were conducted concurrently and independently by 2 reviewers (FA and EA). Having confirmed the same number of results for each database the results were imported into Rayyan systematic review software [35]. Rayan software automatically detects potential duplicates. However, these were only removed after independent confirmation of duplicate status by 2 reviewers (FA and EA). Remaining studies were screened by title and abstract for relevance and eligibility and given a decision outcome of “Include”, “Exclude” or “Maybe” by each reviewer. Reviewers remained blinded to each other’s decision-making. Following un-blinding, disagreements were discussed until consensus for inclusion/exclusion was reached. Due to the high degree of agreement, no third reviewer was necessary for arbitration at this stage.

The remaining full texts were assessed against eligibility criteria and were included in quantitative data synthesis if they met pre-defined inclusion requirements. Concurrently, all bibliographic references from relevant systematic reviews and trials were searched for additional eligible studies.

### 2.4. Assessment of Risk of Bias

An adapted Cochrane Risk of Bias (RoB) [36] tool was piloted and peer-reviewed on an eligible study identified during the initial scoping search [37]. Adjustments were made until the RoB tool captured all domains of bias of interest.

Review authors (FA and EA) independently used the RoB tool to score the studies for the following preselected domains of bias: randomisation, blinding, attrition and reporting bias. Adherence rates and missing outcome data were also reviewed as part of the RoB assessment.

Each RoB outcome was checked against its peer reviewed counterpart and any differences in opinion were mediated by a third reviewer (ER) until agreement was reached. An over-all bias grading of “Low” was assigned if all domains of bias scored low or ≤1 moderate, “Moderate” if ≥2 domains of bias were scored as moderate or “High” if ≥2 domains of bias were scored as high.

Separate to the RoB tool, depression score tools were individually assessed for their respective validity in each population group by investigating prior published validation studies and the method of measuring circulating Omega-3 levels was compared to optimal assessment techniques described in previous work, where specific biomarkers were found to be more suitable in assessing acute changes (plasma phosphatidylcholine EPA and DHA versus platelet and mononuclear cell EPA and DHA for habitual intake) [38,39].

### 2.5. Data Extraction, Management and Synthesis

A piloted data extraction form adapted from the Cochrane Collaboration [40] was used to collect data on study (authors and publication date, trial country, start and end date) and participants’ characteristics (age, sex and comorbidities); intervention (Omega-3 and placebo formulation and dosages, treatment duration and drop-out rates) and outcome measures (circulating Omega 3 levels and depression outcome scale change with side effects/adverse events where available) after being tested on an example eligible study [41]. The form was populated by two independent reviewers. The extracted data were compared, and any inconsistencies corrected by discussion and joint re-extraction of data. Automation tools were not used in the extraction process.

The primary outcome investigated was mean change of depression scores at final available follow up compared to baseline (pre-intervention), and associated *p*-value for significance testing. Where mean change was not available and a predictive equation via multiple time point analysis was used, then this was recorded and reported.

Secondary outcomes such as adverse events, attrition rates and surrogate indicators for adherence (i.e., erythrocyte Omega 3 level changes and pill counts) were reported where available.

From our scoping review and previous systematic reviews of associated topics, we anticipated a heterogeneous set of results, primarily due to variability in assessment tools used to measure severity and establish diagnostic threshold for the different forms of depression. Our intention was to summarise the evidence without meta-analyses in these circumstances.

## 3. Results

The outcome of the search strategy is depicted in the PRISMA flow diagram (see Figure 1). The initial search identified 1551 records from which 212 duplicates were removed. The title and abstracts of 1339 records were screened and studies not meeting inclusion criteria or studies with ineligible design were removed. Of the ten remaining records, three were excluded following full-text assessment for eligibility. Seven studies met full eligibility criteria and have been included in this review [42,43,44,45,46,47]. Bibliographic hand-searching revealed no additional eligible RCTs.

### 3.1. Characteristics of Included Studies

A total of seven trials (See Table 1) reporting on 2575 subjects (1312 in the intervention and 1263 in placebo arms), of which 1903 were male, assessed the effects of long chain Omega-3 supplementation (Ethyl-EPA (E-EPA), EPA and DHA) on depression scores among participants in the Netherlands, United States, Canada, Iran, Taiwan and Germany. Participants recruited had a confirmed diagnosis of a cardiometabolic condition. Two studies recruited participants with diabetes [41,45], one coronary heart disease [46], one cardiovascular disease [43], one chronic heart disease [46], one chronic heart failure [44] and one recent AMI [47]. Five trials recruited participants with pre-existing cardiometabolic and comorbid major depressive disorder. All trials used capsule Omega-3 and placebo. Doses ranged from 1000 to 3000 mg with intervention duration of 10, 12 or 48 weeks. Six out of seven trials used a combination of EPA and DHA at different dosages and ratios while one trial used ethyl EPA (E-EPA). Control groups received placebo capsules of rapeseed, olive, soy, corn oils, a soy/corn blend or edible paraffin. One trial used adjunct sertraline (50 mg) in both treatment arms [42].

### 3.2. Depression Outcomes

One trial used the Montgomery Åsberg Depression Rating Scale (MADRS) [41], four used the Hamilton Depression Rating Scale (HAM-D) [42,43,44,46] and six used the Beck Depression Inventory (BDI-II) [42,43,44,45,46,47] depression scales to report depression outcomes. Where trials were conducted in non-English speaking countries, suitable validated translated questionnaires were used.

Six out of seven trials found no statistically significant improvement versus placebo at the end of intervention period (*p* > 0.05). One trial found a 34% (*p* < 0.05) reduction in BDI-II scores in the intervention group and a Relative Risk Reduction (RRR) of 47.93% (95% CI; 24.89–63.98%) for events (defined as worsened, unchanged or inconsiderably improved depression) versus placebo at the endpoint (Week 10) (see Table 1) [45].

### 3.3. Attrition, Adherence and Adverse Events

Drop-out rates were low among studies and similar between intervention and placebo groups (see Table 1). Reasons for drop-out were unlikely to be related to Omega-3 supplementation, as reasons cited, side effects and adverse events did not significantly differ between intervention and placebo groups (see Table 2). Adherence was measured via assessment of change in Omega-3 blood levels compared to baseline in five trials [41,42,43,44,46]. There were no statistically significant changes in Omega-3 blood levels at endpoint compared to baseline for placebo groups (see Table 1). Omega-3 blood levels in the intervention increased significantly, yet the expected rate of change was not adequately pre-defined, leading to limited confidence in the extent of adherence to the intervention.

### 3.4. Subgroup Analyses

Zimmer et al. and Chang et al. reported results of subgroup analyses for participants with diagnosed depression. In the Taiwanese study, those with very severe depression (*n* = 9, HAM-D score ≥23) benefitted the most with a clinically meaningful −1.74 (95% CI, −3.04 to −0.05, *p* < 0.05) reduction in HAMD scores at 12 weeks [44]. In the German trial, those with depression and not on antidepressants (*n* = 238, BDI-II ≥ 14) did not achieve a significant reduction in depression scores [47]. However, those with depression receiving Omega-3 supplementation and antidepressants (*n* = 33) achieved significant reduction in BDI-II scores compared to those on placebo and antidepressants (*n* = 29) at 48 weeks (Intervention: 20.9 (SD: 7.1) versus Placebo: 24.9 (SD: 8.5), *p* < 0.05), but possible baseline differences were not accounted for as baseline BDI-II scores were not reported.

### 3.5. Risk of Bias

The risk of bias assessment and collated scores for each domain of bias are summarised in Appendix A. Blinding at the outcome analysis stage and selective reporting were predominant issues identified across studies (see Appendix A). Participants and outcome data were accounted for in all trials, and trials scored low for attrition bias, with drop-out rates consistently low across trials.

## 4. Discussion

This systematic review provides a comprehensive summary of the current published clinical trial evidence investigating Omega-3 supplementation for the treatment of depression in cardiometabolic populations. To the best of our knowledge, this is the first review of Omega-3 supplementation for depression in populations with cardiometabolic disease. A recent Cochrane review that investigated Omega-3 for depression in the general adult population was unable to conclude if Omega-3 exerts an antidepressant effect, due to high heterogeneity and very low-quality evidence [48].

There is no current recommendation for dietary Omega-3 intake for adults in disease-free states. There is also lack of agreement on effective Omega-3 supplementation dosage, intervention and duration in the management of depression. Dietary Omega-3 consumption is sporadic and may not provide the optimal strategy for Omega-3 absorption [38,39]. Daily supplementation has been shown to achieve significantly higher Omega-3 incorporation in laboratory measured outcome assessments of platelet and mononuclear cell EPA and DHA, which are biomarkers of habitual intake [38,39]. Observations of people with depression and cardiometabolic disease have revealed significantly lower Omega-3 blood levels compared to the general population [49]. Large biobank and cohort studies indicate that up to a third of adults regularly purchase and consume Omega-3 supplements [50,51,52], yet such products vary greatly in content and composition.

We opted to include only double-blind RCTs in an attempt to reduce bias. Despite this, several trials ranked “Moderate” or “High” for risk of bias, and some lacked the statistical power needed to observe a potential effect. Overall, only one study out of seven noted a significant reduction in depression scores, and the rest did not show superiority over placebo for any duration of treatment. Differences in supplementation formulation, dosage and duration make other conclusions challenging. We confirmed that Omega-3 supplementation, even in relatively high dosages, is safe, with little difference in adverse events or side effects profile compared to placebo.

Current International Society for Nutritional Psychiatry Research Practice (ISNPR) guidelines advocate use of Omega-3s in acceleration (i.e., starting Omega-3 supplementation with an antidepressant at the beginning of the treatment) or augmentation (i.e., supplementation of Omega-3 when an established antidepressant is having an inadequate effect) of antidepressant therapy [53]. Exploratory analysis of those receiving antidepressant therapy and concurrent Omega-3 supplementation did detect a benefit [47], as did a sub-analysis of those with the most severe depression [44]. However, these tests had a small sample size, *n* = 62 and *n* = 9, respectively, and their results should be interpreted with caution. Use of Omega-3 supplementation for acceleration or augmentation of antidepressants was not clarified in the other included trials and no further evidence of significant effect versus placebo was reported for these small subgroups.

The potential synergistic effects of antidepressants and Omega-3 supplementation warrant further investigation. There is an almost complete naivety regarding how these two agents interact, and it is not known whether initiating simultaneous therapy would work better, or what class of antidepressant exerts maximal benefit when paired with Omega-3 supplementation.

### 4.1. Strengths and Limitations of Included Trials

All three depression tools reported in the included studies are widely used and validated depression questionnaires suitable for the population studied. The included studies had a limited justification for supplementary Omega-3 EPA/DHA dosage ratios (where applicable) and intervention duration. Dosages varied considerably from trial to trial and justifications surrounding lower doses of Omega-3 supplementation to prevent side effects appear to be inadequate in the face of evidence demonstrating that adverse events are comparable to placebo [52]. While shifts in Omega-3 blood levels evidenced an effect of Omega-3 supplementation intake, this did not necessarily validate the intervention duration, as there was limited predictability of the degree of increase in Omega-3 blood levels required to observe an effect on depression scores, and time required to see this.

Two studies did not report end-point differences in blood Omega-3 levels compared to placebo or baseline. For those that did, different tests were used and the heterogeneity in these tests used to evidence adherence may have introduced bias, as the validity of some biochemical markers is inferior to others. Indeed, different tests are appropriate depending on the length of intervention prescribed [38,39]. The choice of placebo may not have always been appropriate, as some studies opted for alternative fatty acids such as soybean or olive oil which may have acted as a confounder. Olive oil consumption, for example, has been shown to improve depressive symptomatology in recent clinical trials [54,55].

Few studies reported baseline intake of Omega-3s via diet or supplementation. No study reported Omega-6 intake at baseline, during or at intervention endpoint. This has the potential to confound results, as Omega-6 competes with Omega-3 for absorption. Omega-6 may counteract the anti-inflammatory action of Omega-3, one of the leading theories supporting its potential cardio-protective and mental health benefits [56,57]. Furthermore, concurrent medications used by eligible participants were not reported. This is a significant limitation because medications such as statins, commonly prescribed to people with select cardiometabolic conditions, are known to interact with Omega-3 [58].

Studies also suffered from a low sample size and, while some retained power, others suffered from dropouts, indicating that the lack of a detected effect could have been explained in part to lack of statistical power, not absence of effect.

The largest trial (*n* = 2081) included in this review used a post-test design to evaluate the relationship between Omega-3 supplementation and depression score versus placebo [42]. Self-reported fish consumption was higher in the Omega-3 group at baseline. Furthermore, self-reported fish consumption during the 12-month period did not remain stable, increasing significantly in both intervention and control groups. The post-hoc nature of the analysis, combined with an active placebo (olive oil) and changes in self-reported fish consumption during the course of the study, make any conclusions about the effect of Omega-3 supplementation on depression challenging. This study is illustrative of the challenges facing nutrition research.

### 4.2. Strengths and Limitations of the Review Process

We cannot overlook potential biases occurring during our review process. We looked at several research databases but potentially eligible RCTs may have been missed. However, bibliographic hand-searching of related systematic reviews and RCTs did not reveal any additional trials for inclusion. Additionally, two independent reviewers conducted the searches and study selection, as well as risk of bias assessment and data extraction. Our search was limited to English language only, meaning that published research in non-English journals may not have been detected.

We only included double-blind randomised trials. This allowed us to present the most rigorous published evidence, with the potential of establishing causality, were an effect to be apparent. There was, however, considerate heterogeneity in the diseases reported and depression scores utilised by each included trial. Cardiometabolic disease is an umbrella term describing a series of conditions with shared risk factors, but these conditions can vary significantly from each other. In our review, we included acute cardiometabolic disease (e.g., myocardial infarction) alongside long-term counterparts (e.g., type 2 diabetes). Furthermore, we included studies reporting a variety of different depression scores. These, though validated and appropriate for use, are not interchangeable and can measure different aspects of depression (e.g., severity versus the establishment of a diagnostic threshold).

### 4.3. Quality of the Evidence

We only included double-blind RCTs to limit bias and increase confidence that the effect on depression could be attributable to the intervention with Omega-3 supplementation. However, the risk of bias assessments highlighted concerns with the randomisation, blinding and selective reporting in half of the included trials. Further concerns were identified with select placebos containing components exhibiting potential effects on depression scores. Despite this, publication bias appears to be low and study findings remained broadly consistent across trials.

## 5. Conclusions

Omega-3 supplementation is safe in a population with concurrent cardiometabolic and mental health conditions, but this review cannot confirm if Omega-3 supplementation reduces depression scores in adults with cardiometabolic disease. There are some guidelines recommending Omega-3 supplementation but gold standard evidence from double-blind RCTs suggests that the evidence is currently equivocal. Future targeted studies with longer follow-up are needed to investigate the effect of Omega-3 supplementation on severe depression and clarify interactions with antidepressants.

## Figures and Tables

**Figure 1 nutrients-14-01827-f001:**
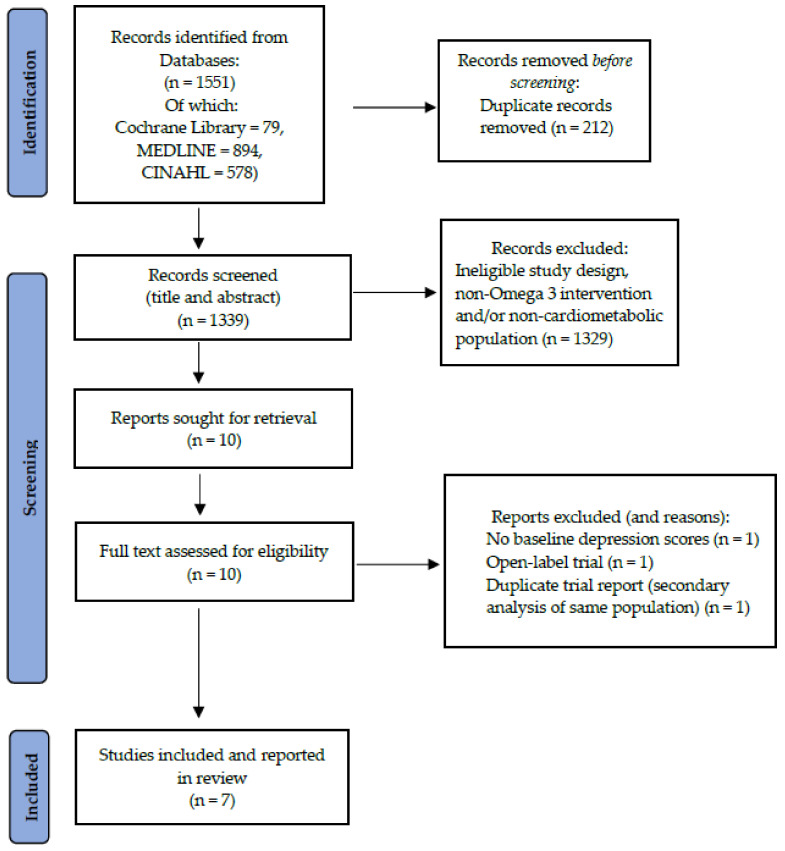
PRISMA flow diagram.

**Table 1 nutrients-14-01827-t001:** Characteristics and outcomes of included studies.

Trial, Start & End Date, Country	Intervention/ Placebo	Sample Size Analysed (Male)	Drop-Out Rates (*n*)	Age in Years (Mean)	Cardiometabolic Conditions	Omega-3 Formulation & Dosage (mg)	Treatment Duration (Weeks)	Circulating Omega-3 Levels Change (Mean)	Depression Outcome Scale	Conclusion
[41] April 2006–July 2007 Netherlands	Intervention	13 (5)	1	53.1 ± 13.8	T1D = 5 T2D = 8 + MDD	1000 mg E-EPA + Antidepressant therapy~	12	Baseline erythrocyte membrane EPA: 0.53% ± 0.17 Week 12: 1.69% ± 0.56	MADRS score at baseline and 12-week follow-up at two-weekly intervals	No improvement versus placebo
	Placebo	12 (7)	2 *	55 ± 8.6	T1D = 5 T2D = 7 + MDD	1000 mg (rapeseed oil + MCT) + Antidepressant therapy~		Baseline erythrocyte membrane EPA: 0.66% ± 0.2 Week 12: 0.61% ± 0.19		
[42] May 2005–December 2008 United States	Intervention	62 (41)	3	58.1 ± 9.4	CHD + MDD	2000 mg (930 mg EPA, 750mg DHA) + 50 mg Sertraline	10	Baseline Omega-3 index (DHA + EPA), % RBC: 4.6 ± 1.5 Week 10: 7.6 ± 1.8	Change at 10 weeks of HAM-D, BDI-II between intervention and placebo	No improvement versus placebo
	Placebo	60 (40)	4	58.6 ± 8.5		2000 mg Corn oil + 50 mg Sertraline		[Omega-3 index (DHA + EPA), % RBC: stable in placebo arm]		
[43] January 2016–March 2017 Taiwan	Intervention	30 (18)	None reported	61.1 ± 9.14	CVD + MDD	3000 mg (2000 mg EPA, 1000 mg DHA)	12	Week 12 total Omega-3 change: 1.96 ± 2.91	Change at 12 weeks of HAMD (and subscale) and BDI between intervention and placebo groups	No improvement versus placebo
	Placebo	29 (20)		61.93 ± 8.95		3000 mg soybean oil		Week 12 total Omega-3 change: −0.58 ± 2.82		
[44] June 2014–May 2016 United States	Intervention	Group 1 = 36 (21) Group 2 = 36 (16)	Group 1 = 1 Group 2 = 3	Group 1 = 57.73 ± 16.14 Group 2 = 58.1 ± 10.16	CHF with NYHA Class ≥II + MDD	Group 1 = 2000 mg (2:1 EPA/DHA) Group 2 = 2000 mg EPA	12	Baseline Omega-3 index (% in RBC): Group 1 = 4.46 ± 1.51 Group 2 = 4.47 ± 0.99 Week 12: Group 1 = 6.79 ± 0.22 Group 2 = 6.32 ± 0.26	Change at 12 weeks of HAMD, BDI-II between intervention and placebo	No improvement versus placebo
	Placebo	36 (13)	4	57.91 ± 11.68		2000 mg Corn oil		Baseline Omega-3 index (% in RBC): 4.51 ± 1.29 Week 12: 4.61 ± 0.26		
[45] July 2014–January 2015 Iran	Intervention	44 (29)	0	51.15 ±7.4	T2D	2700 mg (2:1 EPA/ DHA)	10	Not measured or reported	Difference at 10 weeks in frequency of events (<5-unit decrease inBDI-II-PERSIAN) versus non-events (≥5-unit decrease in BDI-II-PERSIAN)	Favours intervention (p < 0.001)(Relative risk reduction of event: 47.93%, 95% CI: 24.89–63.98%
	Placebo	44 (24)	3	50.56 ± 7.2		3000 mg Edible paraffin				
[46] August 2010–February 2014 Canada	Intervention	45 (35)	5	63.8 ± 9.1	CHD	1900 mg (1200 mg EPA, 600 mg DHA)	12	Baseline: EPA: 26.7 ± 14.2 (μg/mL) DHA: 47.3 ± 20.1 (μg/mL) Week 12: EPA: 41.6 ± 34.4 (μg/mL) DHA: 53.6 ± 29.3 (μg/mL)	Change at 12 weeks of HAMD, BDI-II between intervention and placebo	No improvement versus placebo
	Placebo	47 (35)	5	61.7 ± 8.7		3000 mg 1:1 Soybean/corn oil blend		Baseline: EPA: 28.5 ± 16.6 (μg/mL) DHA: 52.8 ± 23.8 (μg/mL) Week 12: EPA: 23.4 ± 14.4 (μg/mL) DHA: 45.9 ± 20.4 (μg/mL)		
[47] April 2005–June 2007 Germany	Intervention	1046 (802)	None reported	63 † (53, 70)	Post-acute MI admission (3–14 days)	1000 mg (460 mg EPA, 380 mg DHA)	48	Measured but not reported	Change at 48 weeks of BDI-II between intervention and placebo	No improvement versus placebo
	Placebo	1035 (797)		64 † (54, 71)		1000 mg Olive oil				

± Denotes standard deviation. * Excluded due to antidepressant cessation. † Median (lower, upper quartile). ~Comprised of: Tricyclic antidepressants, selective serotonin reuptake inhibitor, noradrenergic and specific serotonergic antidepressant. MADRS: 10-item scale. Each item contains 6 tier statements. Total score range 0–60. Higher scores reflect severity of depressive symptoms. HAMD: 17-item instrument. Total score range. 0–56. Score of 0–7 generally accepted as within normal range. Score of ≥20 indicates at least moderate severity. BDI-II: 21-item instrument. Each item contains 4 (0, 1, 2, 3) tier statements. Total score range 0–63. A score of ≥14 was used as a threshold for the classification of depression. Abbreviations: MCT = Medium-chain Triglycerides, T1D = Type 1 Diabetes, T2D = Type 2 Diabetes, CHD = Coronary Heart Disease as documented by at least 50% stenosis in at least 1 major coronary artery, a history of revascularisation, or hospitalisation for acute coronary syndrome, CVD = Cardiovascular Disease (with either stable myocardial infarction or coronary artery disease), MI = Myocardial Infarction, MDD = Major Depressive Disorder as assessed by the Mini-International Neuropsychiatric Interview (MINI) in accordance with the Diagnostic and Statistical Manual of Mental Disorders 4th edition criteria, RBC = Red Blood Cell (count), CI = Confidence Interval.

**Table 2 nutrients-14-01827-t002:** Side effects and adverse events profile of included studies.

Trial	Group	Reported Side Effects/Adverse Events	Trial Author Conclusions
[41]	Not specified	Belching (*n* = 10), nausea (*n* = 6), diarrhoea (*n* = 5), rash and itching (*n* = 1)	No significant differences in number and type of side effects between Omega-3 and placebo groups
[42]	Intervention	19% (side effects: gastrointestinal complaints, diarrhoea, bloating, or prolonged bleeding) 4 cardiac (2 coronary angioplasty and 4 non-cardiac hospitalisations (worsening heart failure, injury from a fall, kidney stones)	No significant differences in side effects (*p* = 0.72) or adverse events between Omega-3 and placebo groups
	Placebo	22% (side effects: gastrointestinal complaints, diarrhoea, bloating, or prolonged bleeding) 4 cardiac (1 acute MI, 1 coronary angioplasty, 1 cardiac ablation for atrial flutter, 1 implantation of an automatic cardioverter-defibrillator) and 4 non-cardiac hospitalisations (severe influenza, possible allergic reaction to non-study medication and minor accident)	
[43]	Intervention/Placebo	Not reported	No conclusion made
[44]	Intervention	Group 1 (2:1 EPA/DHA): Gastrointestinal discomfort (*n* = 3), nausea (*n* = 4), fishy odour (*n* = 8), upset stomach (*n* = 3), other (*n* = 12) Group 2 (High EPA): Gastrointestinal discomfort (*n* = 2), nausea (*n* = 3), Fishy odour (*n* = 7), Upset stomach (*n* = 1), Other (*n* = 10) (Other: diarrhoea, itching/rash, burping)	No significant differences in side effects or adverse events between Omega-3 (Group 1 or Group 2) and placebo groups (*p* = 0.4)
	Placebo	Gastrointestinal discomfort (*n* = 2), nausea (*n* = 3), Fishy odour (*n* = 4), Upset stomach (*n* = 2), Other (*n* = 4) (Other: diarrhoea, itching/rash, burping)	
[45]	Not specified	Diarrhoea (*n* = 1)	No significant differences in side effects or adverse events between Omega-3 and placebo groups
[46]	Intervention	Pain (*n* = 33) Headache (*n* = 27) Nasopharyngitis (*n* = 37) Upper respiratory tract infection (*n* = 12) Dyspepsia (*n* = 23) Fatigue (*n* = 40) Nausea (*n* = 19) Diarrhoea (*n* = 16) Epigastric discomfort(*n* = 18) Skin eruption (*n* = 15) Itching (*n* = 19) Exanthema (*n* = 8) Eczema (*n* = 8) Increased bruising/bleeding (*n* = 20)	No significant differences in side effects or adverse events between Omega-3 and placebo groups
	Placebo	Pain (*n* = 34) Headache (*n* = 31) Nasopharyngitis (*n* = 34) Upper respiratory tract infection (*n* = 16) Dyspepsia (*n* = 23) Fatigue (*n* = 37) Nausea (*n* = 21) Diarrhoea (*n* = 18 ) Epigastric discomfort(*n* = 21) Skin eruption (*n* = 16) Itching (*n* = 27) Exanthema (*n* = 12) Eczema (*n* = 11) Increased bruising/bleeding (*n* = 21)	
[47]	Intervention	Neoplasms (*n* = 19) Cardiac device therapeutic procedures (*n* = 16) Malignancies (*n* = 32) Rhythmologic events (*n* = 99)	No significant differences in total number of side effects or adverse events between Omega-3 (1769 events) and placebo (1804 events) groups (*p* = 0.27) Note: Side effects/adverse events of main study (*n* = 3851) reported.
	Placebo	Neoplasms (*n* = 8) Cardiac device therapeutic procedures (*n* = 2) Malignancies (*n* = 26) Rhythmologic events (*n* = 84)	

## Data Availability

Data for each included study reported in this review may be obtained by referring to the respective publication.

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
