# Peer review of "The Effects of Omega-3 Supplementation on Depression in Adults with Cardiometabolic Disease: A Systematic Review of Randomised Control Trials"

_nutrients, 2022, doi:10.3390/nu14091827_

Round 1

Reviewer 1 Report

This is an appropriately conducted SLR, that followed the PROSPERO registered protocol and reported as per PRISMA. Findings are important as the current evidence from primary study is conflicting. I have provided a few suggestions below:

  1. The author mentioned the primary endpoint in the abstract please mention the secondary endpoints well, as they are mentioned in the method section.
  2. Also please mention the "Primary" endpoint in the method section as well.
  3. Please put the full form of RRR in the abstract at the first occurrence.
  4. For line 54, is there a systematic review on "use of antidepressants with increased risk of major adverse cardiovascular events" if yes please use that to establish this point.
  5. Please look at line 102, seems some editing error "cardiometabolic disease.3.Results"
  6. Please check the numbering of the Table. There is only Table 2 in the manuscript. Seems one table is missing.
  7. Please use terms "Omega-3s" and "Omega-3 supplements" uniformally across
  8. Pleas revisit text such as "po-tential" in line 331 "suf-fered" in line 338. this is observed at other places as well.
  9. Correct the title of Figure 1, "Figure 1. PRISMA flow diagra."
  10. I see Zimmer et al study is an unexpected large study, if there would have been a possiblity of meta-analysis this study would have had maximum impact on the pooled estimates. Hence, author can look to describe this trial in more detail in discussion section even if they haven't done meta-analysis.

Reviewer 2 Report

1) The aim of the study is unclear. There is no information on the effects of Omega-3 supplements on depression in “Introduction”, while the authors conclude the ineffectiveness of Omega-3 in patients with cardiometabolic diseases and depression. One may think the authors had interested in the effects of Omega-3 on cardiometabolic diseases in patients with mental diseases.

2) Are there any double-blind RDTs showing a significant reduction in depression scores, and superior to placebo during treatment, except for Ref.40?

3) Table 1 is missing

4) There are many words with hyphens inserted.

5) Lines 102-103 should be checked.

Round 2

Reviewer 2 Report

The authors have addressed my concerns.